# The Cross-Dike Failure Probability by Wave Overtopping over Grass-Covered and Damaged Dikes

**Vera M. van Bergeijk** [1,*], **Vincent A. Verdonk** [1,2], **Jord J. Warmink** [1] and **Suzanne J. M. H. Hulscher** [1]

1   Department of Marine and Fluvial Systems, University of Twente, Drienerlolaan 5,
    7522 NB Enschede, The Netherlands; vincent.verdonk@anteagroup.com (V.A.V.);
    j.j.warmink@utwente.nl (J.J.W.); s.j.m.h.hulscher@utwente.nl (S.J.M.H.H.)
2   Antea Group, Tolhuisweg 57, 8443 DV Heerenveen, The Netherlands
*   Correspondence: v.m.vanbergeijk@utwente.nl

**Abstract:** A probabilistic framework is developed to calculate the cross-dike failure probability by overtopping waves on grass-covered dikes. The cross-dike failure probability of dike profiles including transitions and damages can be computed to find the most likely location of failure and quantify the decrease in the failure probability when this location is strengthened. The erosion depth along the dike profile is calculated using probability distributions for the water level, wind speed and dike cover strength. Failure is defined as the exceedance of 20 cm erosion depth when the topsoil of the grass cover is eroded. The cross-dike failure probability shows that the landward toe is the most vulnerable location for wave overtopping. Herein, the quality of the grass cover significantly affects the failure probability up to a factor 1000. Next, the failure probability for different types of damages on the landward slope are calculated. In case of a damage where the grass cover is still intact and strong, the dike is most likely to fail at the landward toe due to high flow velocity and additional load due to the slope change. However, when the grass cover is also damaged, the probability of failure at the damage is between 4 and 125 times higher than for a regular dike profile.

**Keywords:** wave overtopping; erosion; levee; cover; probabilistic framework

## 1. Introduction

Coastal and fluvial areas are threatened by flooding by seas or rivers during a storm. Earthen dikes with a grass cover on top protect the hinterland against flooding and are one of the main flood defence structures in the Netherlands. Earthen dikes are also found among others in Western Europe [1], USA [2] and China [3]. However, many of these dikes need to be strengthened due to sea level rise and increase in peak river discharges as a result of climate change. This asks for cost-effective design solutions and accurate assessment tools.

Wave overtopping is one of the main reasons causing failure of grass-covered dikes. During a storm, high waves can overtop the dike although the water level is below the dike height and flow down the landward slope with significant erosive action [4,5]. The grass cover and the clay layer with the grass roots (Figure 1a) exert a crucial role in protecting earthen dikes from erosion [6–9]. The erosion resistance of the dike cover is determined by the topsoil, defined as the upper 20 cm of the grass cover (Figure 2), including the type of vegetation and its root system. The erosion resistance decreases in the subsoil layer underneath the topsoil where less roots are present; here, the erosion resistance is determined by the clay quality [4,10].

The probability of failure by wave overtopping is usually calculated based on the average overtopping discharge [11–13]. Failure is defined as the exceedance of a maximum allowable overtopping discharge that varies between 0.1 and 10 L/s/m. There are two disadvantages of using the overtopping discharge to describe failure. Firstly, the relation between the overtopping discharge and the resulting grass cover erosion is unclear. For example in the

Netherlands, failure is defined as an exceedance of 20 cm erosion depth [14]. An erosion depth of 20 cm can be caused by a wide range of overtopping discharges depending on hydraulic conditions and the strength of the dike cover [15,16]. Secondly, the methods based on the average overtopping discharge do not include a cross-dike component. Therefore, it is unknown where exactly the dike is most likely to fail.

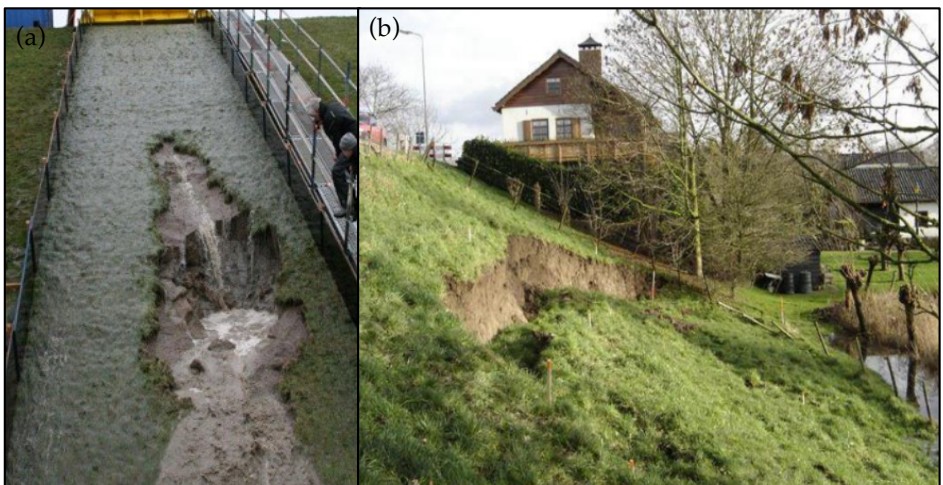

**Figure 1.** (**a**) Dike cover erosion on the landward slope and toe during wave overtopping field tests in the Netherlands (Photo by Juan Pablo Aguilar Lopez). (**b**) A slope instability of a grass-covered clay dike in the Netherlands [17].

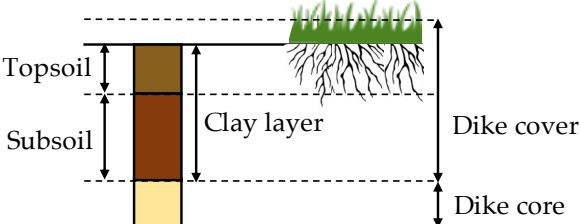

**Figure 2.** Schematization of the dike cover where the grass roots strengthen the topsoil of approximate 20 cm and the clay quality determines the strength of the subsoil.

Erosion models for wave overtopping are introduced to overcome these problems, for example the Cumulative Overload Method (COM) [18] or the analytical Grass-Erosion Model (GEM) [16,19]. The COM is based on the flow velocity and the cover strength is included in the model using a critical velocity. The damage number can be computed at multiple locations using influence factors for the flow acceleration and the effect of transitions [20]. The damage number for failure is empirically determined for the erosion of the topsoil, but the damage number for larger erosion depths is unknown. The GEM calculates the flow velocity and erosion depth along the dike profile and can easily be extended to larger erosion depths. The variation in flow velocity is calculated using the analytical formulas of Van Bergeijk et al. [21] and the effect of transitions can be included using a turbulence parameter. Although both methods are promising, they have solely been applied to compute failure during a storm and not yet for failure probabilities.

The cross-dike failure probability is computed by Aguilar-López et al. [15] for individual overtopping waves during a storm. This enables identification of the locations that are most likely to fail and might need additional strengthening measures. In the study of Aguilar-López et al. [15], Hoffmans' erosion model [4] is coupled to the numerical hydrodynamic model of Bomers et al. [22] to compute the erosion depth along the dike profile. However, due to high computational costs of the numerical model, probabilistic computations are only possible using an emulator [15]. This emulator was trained for a

specific dike profile and needs to be trained again for other dike configurations. Moreover, the emulator is a black box and the link to the physical variables is not clear. Therefore, it is difficult to understand what the important variables are for dike cover failure. For example, the load is solely described by the average overtopping discharge that depends on many different parameters so the effect of the water level or wave height cannot be determined using this method.

A new framework for the failure probability by wave overtopping is developed based on the GEM [16]. This model couples the analytical formulas for the maximum flow velocity of Van Bergeijk et al. [21] with the erosion model of Hoffmans [4] to compute the erosion depth along the dike profile. The GEM has been used to compute the erosion depth of the upper cover layer during a storm [16], but needs to be extended for the computation of failure probabilities. The advantage of an analytical model compared to a numerical model is that analytical models are fast, thus failure probabilities can be computed without an emulator. Moreover, the analytical formulas contain physical parameters related to bottom friction, turbulence and cover strength, and therefore it is possible to find the effect of these physical variables on the failure probability. This new framework can be applied to multiple dike configurations, since the dike geometry and the hydraulic load are the only required boundary conditions.

Importantly, this framework can account for damages in the dike cover to obtain insights into the residual dike strength. Residual dike strength is defined as the ability of the flood defence to continue its water retaining function after it has failed according to the failure definition [23]. For wave overtopping, the residual dike strength is characterised by the clay cover underneath the grass cover once the topsoil is eroded. This clay layer still protects the dike core for overtopping and needs to be eroded completely before the dike breaches [9] (Figure 1a). However, more knowledge is required on how fast the erosion progresses once the topsoil is eroded before the residual strength of the clay layer can be considered and the failure definition for overtopping can be extended to larger erosion depths.

Moreover, the interaction between different failure mechanisms needs to be considered for some cases of residual dike strength. For example, a small slope instability does not result in flooding (Figure 1b) and progressive slope instabilities are necessary before the dike loses its water retaining function [23,24]. The effect of a small slope instability on other failure mechanisms needs to be determined before progressive slope instabilities can be included in safety assessments. A dike with a small slope instability is more vulnerable for wave overtopping due to the damaged dike cover and the formation of a vertical cliff at the instability that affects the hydraulic load [17]. The effect of a slope instability on the failure probability by wave overtopping is unknown.

The goal of this study is to calculate the effect of damages in the dike profile on the failure probability by wave overtopping using a new probabilistic framework. The main innovative component of the study is that the effect of transitions and damages on the failure probability can be computed to quantify the increase in failure probability caused by these weak spots in the dike profile. The landward toe is used as an example of a transitions in this study, where the slope change leads to an additional load on the cover. Additionally, the framework is applied to damaged dike profiles, such as a dike profile with an erosion hole or a small slope instability. These damages have two effects on the erosion by overtopping waves: (1) the load increases due to jet impact landward of the vertical cliff, and (2) the cover strength is reduced near the damage. Quantification of the failure probability at transitions and damages can help to improve local dike strengthening measures and maintenance strategies.

## 2. Framework for the Failure Probability by Wave Overtopping

A probabilistic framework is developed to calculate the failure probability by wave overtopping along a grass-covered dike crest and landward slope. First, the failure probability conditional to the water level $P_{f|h}$ is calculated (Figure 3). The hydraulic load

distribution at the start of the dike crest consists of all overtopping wave volumes during the storm and depends on the water level $h$, the wind speed $u_{10}$, the fetch length and the geometry of the outer slope. The other required input variables are the dike geometry and the dike cover strength described by the critical velocity $U_C$. The GEM calculates the erosion depth along the dike profile of every overtopping wave based on the overtopping volume and the dike characteristics. The total erosion depth during the storm $d(x)$ is calculated at every cross-dike coordinate $x$ by a summation of all overtopping waves.

The conditional failure probability is determined from the limit state function $Z$ that expresses the difference between the strength and the load. The strength of dike cover is set to the upper cover layer of 20 cm and the load is described $d(x)$. Failure is defined as $Z < 0$ which happens when the erosion depth exceeds 20 cm according to the Dutch failure definition [14]. The probability of failure corresponds to the probability $P(Z < 0)$. A Monte Carlo analysis with $2 \times 10^4$ samples (see Appendix A for convergence of $P_f$) is performed where each sample corresponds to a storm event and therefore the failure probability in this study is the failure probability per storm event. The wind speed and critical velocity are sampled from their distribution for every storm event while keeping the water level constant, resulting in the failure probability conditional to the water level $P_{f|h}$.

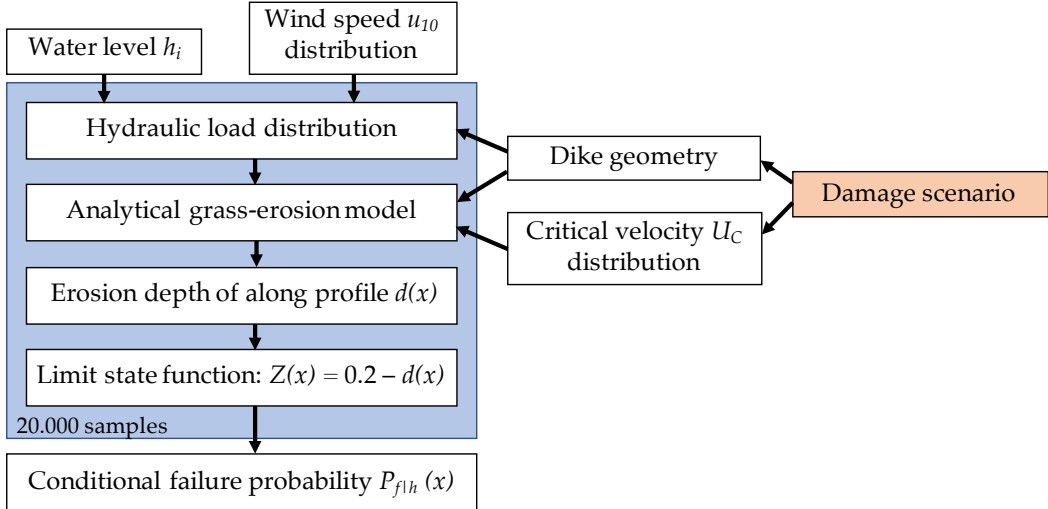

**Figure 3.** Schematization of the framework to calculate the failure probability conditional to the water level $P_{f|h}$.

The conditional failure probability $P_{f|h}$ is computed for several water levels to construct a fragility curve as illustrated in Figure 4. The conditional failure probabilities are significant for high water levels with a low probability. The probability of failure $P_f$ is calculated by numerical integration of the conditional failure probability and the probability density function of the water level $f(h)$

$$P_f(x) = \int P_{f|h}(x) f(h) dh \tag{1}$$

The framework can be applied to damaged dike profiles such as an erosion hole or a slope instability. These damages influence the dike geometry and the strength of dike cover as indicated by the orange box in Figure 3.

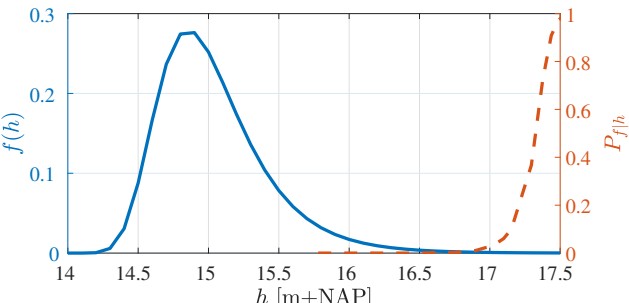

**Figure 4.** Example of the probability density function of the water level $f(h)$ (solid line) and a fragility curve (dashed line) showing the conditional failure probability $P_{f|h}$ as function of the water level $h$.

### 2.1. Hydraulic Load

The sampled water level and sampled wind speed are used together with the fetch length to calculate the significant wave height $H_s$ and the wave period $T_p$ at the outer toe of the dike using the Bretschneider equations. The return frequencies for the water levels are obtained from the Hydra-NL WBI 2017 software which includes the water levels and their return periods for every river dike section in the Netherlands based on an uncertainty analysis of numerical river models [25]. The wind statistics of Caires [26] are used to obtain the cumulative probability distribution for the wind speed $u_{10}$ at a standard landscape roughness and a standard height of 10 m.

The storm duration was fixed for all simulations and set to 6 h based on previous research [17] and Dutch assessment requirements [27]. The 2% exceedance run-up height, average overtopping discharge and number of overtopping waves during the storm are calculated using the formulas in the EurOtop Manual [11]. These variables are used to generate the individual overtopping volumes during the storm according to the probability exceedance distribution as described by Van Bergeijk et al. [16]. The method for the hydraulic load is described in more detail in Verdonk [28].

### 2.2. Analytical Grass-Erosion Model

The erosion depth along the dike profile is calculated using the analytical grass-erosion model GEM where the analytical formulas for the overtopping flow velocities of Van Bergeijk et al. [21] are coupled to the erosion model of Hoffmans [4] that has been adapted to account for cross-dike variations in the load and strength so the effect of transitions and damages can be included [16,19]. Hoffmans' erosion model [4] describes the scour erosion of clay and grass covers by overtopping waves based on an erosion model for jet scour [29–32]. Scour erosion is the result of high flow velocities and locally increased turbulence, and therefore Jorissen and Vrijling [32] introduced a turbulence parameter $\omega$ to account for the effect of turbulence on the scour erosion. This turbulence parameter is related to the depth-averaged relative turbulence intensity $r_0$ as [7,32,33]

$$\omega = 1.5 + 5 \cdot r_0 \tag{2}$$

For wave overtopping flow, the depth-averaged relative turbulence intensity $r_0$ relates to the friction of the bed (see Appendix B for this relation) and the turbulence parameter in the erosion model accounts for the increase in the hydraulic load as the result of turbulence generated by bed friction [4,22]. Hoffmans et al. [7] estimated a range for the turbulence parameter on the slope ($\omega_{slope} = 2.00 - 2.75$) for wave overtopping on grass-covered dikes (Appendix B).

The total erosion depth during a storm event is calculated by first computing the flow velocity $U$ along the dike profile of every overtopping wave $i$ using analytical flow formulas that were derived from the 1D shallow water equations [21]. The overtopping volume is used as a boundary condition and the maximum flow velocity along the dike profile depends on the cover type and the dike geometry including the slope angle. Next, the total

erosion depth along the dike profile $d(x)$ is calculated by summing over all overtopping waves $N$.

$$d(x) = \sum_i^N \left( \omega^2(x) U_i^2(x) - U_t^2 \right) T_0 C_E \quad \text{for } \omega(x) U_i(x) > U_t \qquad (3)$$

with the threshold flow velocity $U_t$, the overtopping period $T_0$ and the inverse cover strength parameter $C_E$. Erosion of the grass cover starts once the hydraulic load-described by turbulence parameter and flow velocity ($\omega U$)–exceeds the threshold flow velocity which is a factor 2.4 larger than the critical velocity $U_C$ (Table 1). The turbulence parameter depends on the location along the dike profile based on three values for the crest, landward slope and landward toe (Table 1)

$$\omega(x) = \begin{cases} 2.35, & \text{Crest} \\ 2.00, & \text{Landward slope} \\ 2.75, & \text{Landward toe} \end{cases} \qquad (4)$$

**Table 1.** The turbulence parameter $\omega$ and the threshold velocity $U_t$ in the GEM are determined from calibration or measurements of wave overtopping field tests on grass-covered dikes in the Netherlands and Belgium.

| Parameter | Relation | Method |
|---|---|---|
| Turbulence parameter | $\omega_{toe} = 2.75$ | Calibration using the measured erosion depth at 7 field tests [19,34] |
| | $\omega_{crest} = 2.00$ | Determined from measured pressure fluctuations at Millingen a/d Rijn [35] |
| | $\omega_{slope} = 2.35$ | Determined from measured pressure fluctuations at Millingen a/d Rijn [35] |
| Threshold velocity | $U_t = 2.4 U_C$ | Calibration using the measured erosion depth at 7 field tests [19,34] |

The critical velocity and the inverse strength parameter depend on the quality of the dike cover which consists of grass or clay (Table 2). The uncertainty in the cover strength is simulated using a log-normal distribution for the critical velocity where the mean and the coefficient of variation are based on Aguilar-López et al. [15]. The inverse strength parameter is related to the erosion speed, which is larger for a poor grass cover compared to a good grass cover [36]. Once the topsoil of the grass cover is eroded (Figure 2), the clay layer underneath contains only a small amount of roots and the strength of the cover mainly depends on the cohesion of clay. The lower resistance against erosion results in a smaller critical velocity and a larger inverse strength parameter for clay compared to grass.

The GEM is the only model for the erosion of the topsoil layer by overtopping waves that takes cross-dike variations in load into account and can therefore be applied to transitions and damages. The GEM has been validated for a storm event by Van Bergeijk et al. [16] where they show that the model is able to accurately predict the erosion depth along the dike profile measured during wave overtopping field tests in the Netherlands and Belgium. Moreover, the erosion model and the grass-cover distributions in the framework have been validated by Bomers et al. [22] and Aguilar-López et al. [15].

**Table 2.** The critical velocity $U_C$ and the inverse cover strength parameter $C_E$ of the dike cover for three grass qualities and a good clay quality together with the coefficient of variation $CV$ used for the distribution of the critical velocity.

| | | Grass | | | Clay | |
|---|---|---|---|---|---|---|
| | | Good | Average | Poor | Good | Source |
| $U_C$ | [m/s] | 6.5 | 4 | 2.5 | 0.85 | Aguilar-López et al. [15] |
| $CV$ | [-] | 0.3 | 0.3 | 0.3 | 0.1 | Aguilar-López et al. [15] |
| $C_E$ | [s/m] | $1 \times 10^{-6}$ | $2 \times 10^{-6}$ | $3 \times 10^{-6}$ | $50 \times 10^{-6}$ | Verheij et al. [36] |

## 3. Methods

This framework is applied to a grass-covered river dike in the Netherlands. Firstly, the cross-dike failure probability is computed to identify vulnerable locations for wave overtopping failure. Secondly, a relation for the additional load due to jet impact near damages is derived and the framework is applied to damaged dike profiles. The failure probabilities for damaged spots are computed for a varying cover quality to simulate different types of damages. These failure probabilities of damaged spots are compared to the failure probability of the landward toe to determine the effects of damaged dike profiles on the failure probability.

### 3.1. Study Area

The framework is applied to a dike near Millingen a/d Rijn in the Netherlands close to the junction of the Rhine, the Pannerdensch canal, and the Waal (Figure 5a). This is a grass-covered dike with homogeneous clay core [22]. At this location, wave overtopping tests have been performed in 2013 to determine the erosion resistance of the grass cover for wave overtopping and the influence of an asphalt road on the erosion [37]. Aguilar-López et al. [15] used the same dike section enabling comparison of methods and failure probabilities. Additionally, the turbulence parameter was determined during the overtopping tests from the measured pressure fluctuations (Table 1).

The dike height of 17.93 m+NAP was determined using AHN viewer [38]. The dike geometry is characterised by an outer bed level of 9.4 m+NAP, a crest width of 4.20 m and a horizontal slope length of 17.20 m. Both the water side and landward slope have a steepness of $\cot(\varphi) = 3$ (Figure 5b). A friction coefficient $f$ of 0.01 was used in the GEM for the grass cover. A smooth waterside slope was assumed with a reduction factor $\gamma_f = 1$ for the run-up and overtopping equations [11].

The wind statistics of Caires [26] at the measurement station Deelden are used for this study area. Multiple simulations with different wind directions and their corresponding fetch length showed that the western wind direction (270 degree) with an effective fetch length of 2.785 km was dominant for wave overtopping [28].

The conditional failure probabilities are computed for 24 water levels varying between 15.78 m+NAP with a return period of 100 years and 17.64 m+NAP with a return period of $10^6$ years, which is the maximum return period in the Hydra-NL software. The maximum water level of 17.64 m+NAP means that we do not compute the conditional failure probabilities for freeboards smaller than 0.3 m. However, this will not affect the total failure probability since the exceedance probabilities of these water levels are negligible for this location.

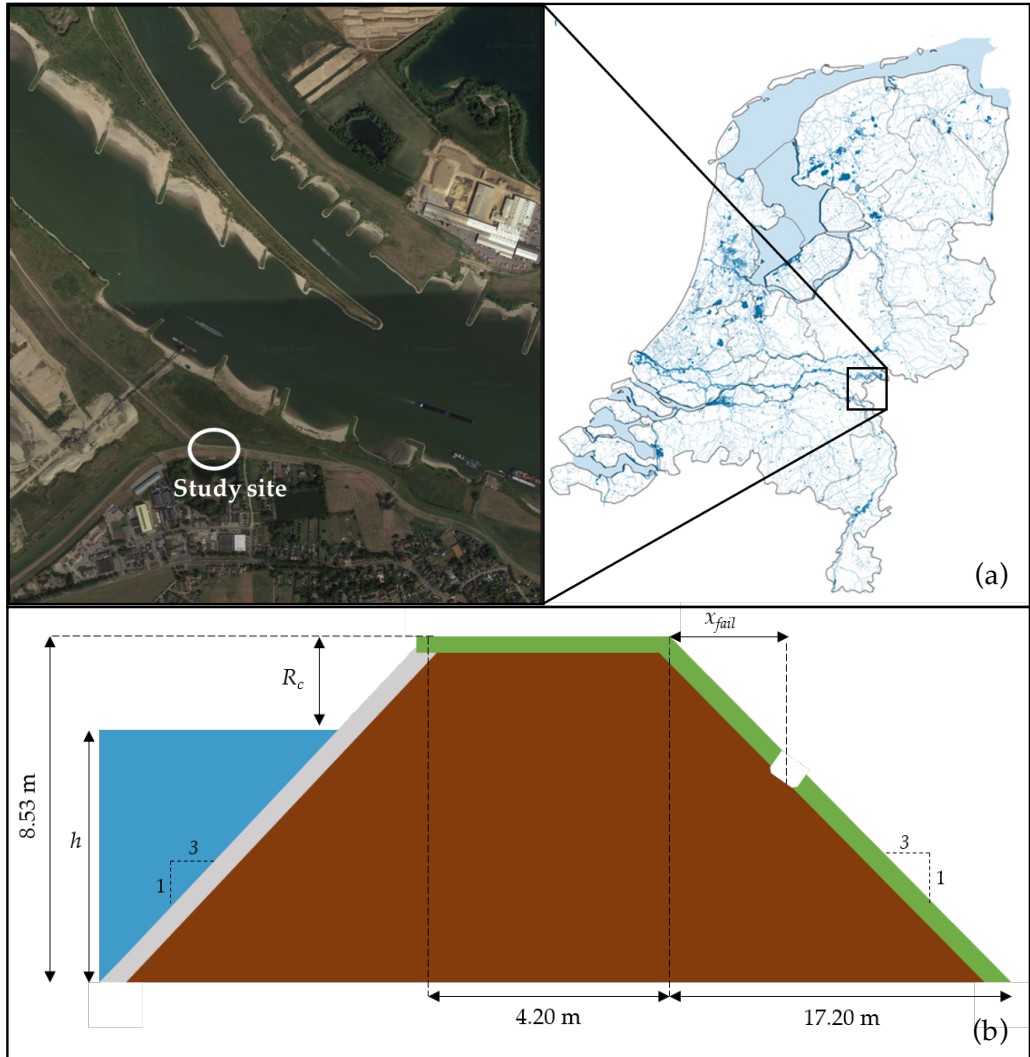

**Figure 5.** (**a**) Top view of the study site at Millingen a/d Rijn and the location in the Netherlands. Retrieved from Google Earth, earth.google.com and Oppervlaktewater in Nederland, www.clo.nl. (accessed on 13 November 2020). (**b**) The dike geometry with a smooth waterside slope and a grass-covered crest and landward slope including the water level $h$, the free crest height $R_c$ and the location of failure $x_{fail}$.

### 3.2. Cross-Dike Failure Probability

The failure probability $P_f$ is calculated along the dike profile every 0.5 m including the end of the crest and the landward slope to save computational costs. This spatial step of 0.5 m has no effect on the results since the analytical formulas in the framework are independent of the spatial step [21]. The 24 conditional failure probabilities $P_{f|h}$ are computed at every cross-dike location and numerically integrated using Equation (1) to obtain the cross-dike failure probability $P_f(x)$.

The effect of the grass cover quality on the failure probability is investigated by computing $P_f(x)$ for three grass qualities: poor, average and good (Table 2). The turbulence parameter $\omega$ is kept constant along the profile to solely identify the effect of the grass cover quality, where a value of $\omega = 2.00$ is used corresponding to the measured turbulence parameter on the slope (Table 1).

Next, $P_f(x)$ is also computed for a variation in the turbulence parameter along the dike profile as described by Equation (4) using an average grass quality. Comparison between the failure probability for a constant turbulence parameter and a varying turbulence parameter quantifies the underestimation in the failure probability when the turbulence parameter is not locally adapted for transitions.

### 3.3. Additional Load at Damaged Spots

A vertical cliff forms at damages on the landward slope such as an existing erosion hole or a slope instability [17,39] (Figures 1b and 6b). When the overtopping wave flows over this vertical cliff, a jet will form that impacts below the cliff (Figure 6a). The jet reattaches to the dike slope at a small distance landward of the cliff where the load on the cover increases due to the impact and a local increase in turbulence [4,40,41]. This additional load has not been quantified for wave overtopping erosion models.

The additional load on the cover is often expressed as the energy dissipation of the flow indicated by a friction factor, for example in the case of high-velocity air-water flows over a grass cover [42] or flow over stepped spillways [43]. The effect of bed friction is included in the GEM in the turbulence parameter, which originates from the local scour parameter that accounts for the effect of local turbulence on scour erosion created by both submerged and plunging jets [4,32,33]. The flow over a cliff at the damage shows similarities with both the flow over stepped spillways and a plunging jet and therefore the additional load by damages is included in the GEM using the turbulence parameter. The value of the turbulence parameter $\omega$ for overtopping flow over damaged spots is unknown and needs to be determined for this study. Since the energy dissipation depends on the cover type [42], we assume that the turbulence parameter for the damages depends on the critical velocity which is used in the model to simulate different cover types.

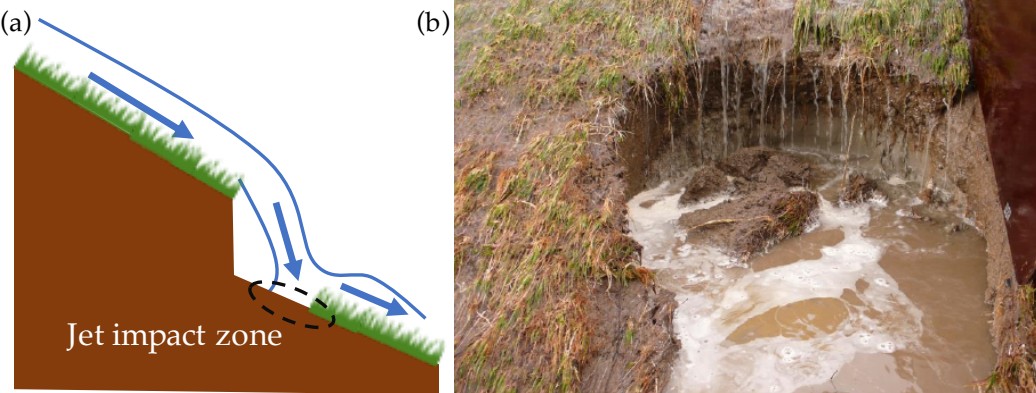

**Figure 6.** (**a**) Small cliff at a damaged spot leads to to a jet forming that impacts in the jet impact zone (white circle) resulting in an additional load. (**b**) Photo of damage during the overtopping test at St. Philipsland resulting in a small cliff [44].

The turbulence parameter for damaged spots is determined from the results of the wave overtopping tests on real grass-covered dikes in the Netherlands and Belgium in 2008–2012 (Table 3). During these tests, the dike cover eroded most at the landward toe due to high flow velocities [16], on the slope near weak spots such as molehills and bald spots [20] or on the upper slope near the crest due to wave impact [41]. The erosion in the latter two cases resulted in damages with a small cliff on the slope that led to failure of the dike cover (Figure 6b). Eight test sections (Table 3) were selected where the cover erosion on the slope resulted in the formation of a vertical cliff with a maximum depth of 20 cm. These damages to the grass cover were the result of animal burrows such as mice, moles and rabbits (Kattendijke 2, Wijmeers 1, Tielrodebroek 1 and 2) or an erosion hole formed by gradual scour erosion (Afsluidijk 2 and Wijmeers 3), bulging (Tholen 3) or roll-up (St. Philipsland). The test conditions and results are used to calibrate the turbulence parameter for small damages on the slope.

**Table 3.** The location of failure $x_{fail}$ measured from the start of the landward slope for the eight test sections in the Netherlands and Belgium. The test conditions resulting in failure are described by the critical velocity $U_C$, the average overtopping discharge $q$ and the simulated storm duration $t_{storm}$. The fractions for $q$ indicates that the cover failed at a fraction of the storm duration for that specific discharge.

| Test Section | $x_{fail}$ [m] | $U_C$ [m/s] | $q$ [L/s/m] | $t_{storm}$ [h] | Source |
|:---:|:---:|:---:|:---:|:---:|:---:|
| Afsluitdijk 2 | 2.8 | 4.0 | 1, 10 | 6 | Bakker et al. [45] |
| Tielrodebroek 1 | 1.9 | 1.2 | 1, 10, $30(\frac{1}{3})$ | 2 | Peeters et al. [46] |
| Tielrodebroek 2 | 1.9 | 1.6 | 1, 10, $30(\frac{1}{6})$ | 2 | Peeters et al. [46] |
| Wijmeers 1 | 1.7 | 3.5 | 1, 5, 25 | 2 | Pleijter et al. [47] |
| Wijmeers 3 | 1.3 | 3.0 | 25 | 2 | Pleijter et al. [47] |
| Kattendijke 2 | 6.6 | 6.5 | 30, 50 | 6 | Bakker et al. [44] |
| St. Philipsland | 6.5 | 6.5 | 0.1, 1, 10, 30, 50 | 6 | Bakker et al. [44] |
| Tholen 3 | 6.5 | 0.0 | 1, 5 $(\frac{2}{3})$ | 6 | Bakker et al. [48] |

Firstly, the location of failure $x_{fail}$ (Figure 5b) and the critical velocity of the test section are obtained from the reports (Table 3). Next, the test conditions are simulated in the GEM using the average overtopping discharge and the storm duration to find the combination of the turbulence parameter and the critical velocity that lead to failure ($d(x_{fail}) = 20$ cm) for each test section (Figure 7). This results in eight failure points for the eight test sections. A relationship between the turbulence parameters and the critical velocity is determined from a linear fit through these failure points. The inverse strength parameter depends on the cover type (Table 2) and during the calibration the cover types are distinguished as poor ($U_C \leq 3$ m/s), average ($3$ m/s $< U_C \leq 5$ m/s) and good ($U_C > 5$ m/s).

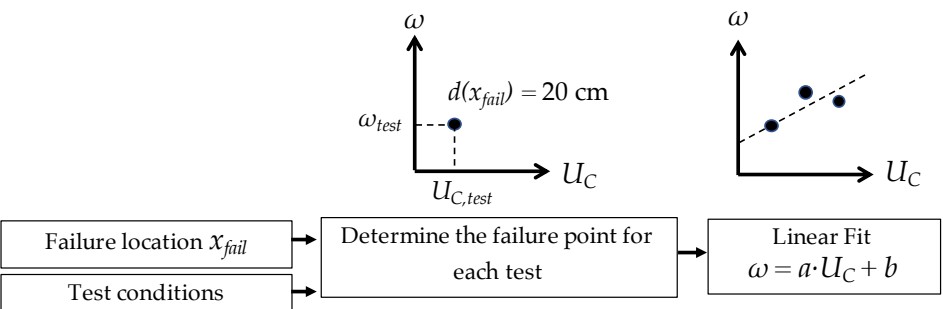

**Figure 7.** Schematization of the method for the calibration of the turbulence parameter $\omega$ at damaged spots as a function of the critical velocity $U_C$ using the analytical grass-erosion model (GEM) and the overtopping tests.

*3.4. Failure Probability of Damages on the Landward Slope*

Damages on the landward slope have two effects on the cover erosion by overtopping waves. Firstly, a vertical cliff forms at the damaged spot leading to an additional load on the grass cover. This effect is included in the model using the turbulence parameter as discussed in the previous section. Secondly, the dike cover strength is reduced near the damage leading to a poorer grass quality or a bare clay cover in case of an erosion hole. The reduction in cover strength is modelled in the analytical grass model by changing the critical flow velocity and the inverse strength parameter from a good grass quality to an average or poor quality, or to clay for a damage where the grass cover is eroded completely (Table 2).

The failure probability for damaged spots $P_{f,damage}(x)$ is computed along the landward slope for four dike covers: bare clay, poor grass, average grass and good grass. The distributions for the critical velocity and the value of the inverse strength parameter in Table 2 are used to simulate the dike cover type. The mean critical velocity for

each cover type is used to calculate the turbulence parameter near the damage using the calibrated relationship.

The ratio $\Delta P_f(x)$ is used to quantify the difference in failure probability for a undamaged and damaged dike profiles

$$\Delta P_f(x) = P_{f,damage}(x)/P_{f,toe} \tag{5}$$

with the failure probability at the landward toe $P_{f,toe}$. The landward toe is the location where a regular dike profile without any weak spots in the dike cover will fail due to high flow velocities at the end of the slope and additional load due to the slope change. The regular dike profile is modelled assuming a good grass cover and a turbulence parameter of 2.75 for the additional load at the landward toe (Table 1). In cases where the ratio is smaller than 1, the dike is most likely to fail at the landward toe and the damage is not the weakest location. Contrary, in cases where the ratio is larger than 1, the damaged spot is most likely to fail for overtopping.

## 4. Results

### 4.1. Cross-Dike Failure Probability

The cross-dike failure probability follows the variation in flow velocity along the profile and is maximum at the end of the slope (Figure 8). The flow velocity decreases over the crest ($x = 0 - 4.2$ m) due to bottom friction, increases over the slope ($x = 4.2 - 21.4$ m) until a balance is reached between the gravitational acceleration and the bottom friction followed by a decrease after the landward toe ($x > 21.4$ m) due to bottom friction. The same cross-dike variation is observed in the failure probability for a constant turbulence parameter, because the load variation in this case is solely determined by the flow velocity.

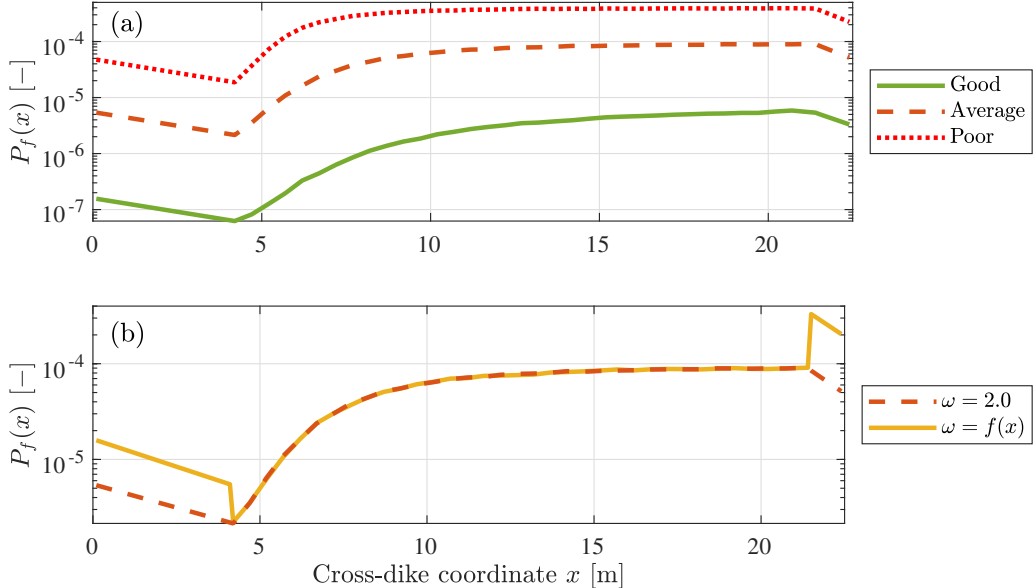

**Figure 8.** (**a**) The cross-dike failure probability $P_f(x)$ for three grass qualities and a constant turbulence parameter $\omega$. (**b**) The cross-dike failure probability $P_f(x)$ for an average grass quality with a constant turbulence parameter $\omega$ and the turbulence parameter of Equation (4).

The grass quality has a large effect on the failure probability. The failure probability decreases with a factor up to $10^2$ when the grass cover quality increases from average to good and increases with a factor 10 when the grass quality decreases from average to poor (Figure 8a). The difference between the failure probabilities is maximal at the end of the crest at a cross-dike distance $x$ of 4.2 m. The turbulence parameter can locally increase the failure probability as can be seen on the crest and at the landward toe in Figure 8b. The failure probability for a varying turbulence parameter (that is $\omega(x)$ in Equation (4)) no

longer follows the variation in flow velocity but increases at locations where the turbulence parameter is higher.

In all cases, the landward toe has the highest probability of failure. For an average grass quality, the landward toe is 35 times more likely to fail compared to the upper slope. This increases to a factor 150 when the turbulence parameter for the landward toe is used to describe the additional load due the change in slope. The failure probability at the landward toe increases from $9.1 \times 10^{-5}$ for a constant turbulence parameter to $3.3 \times 10^{-4}$ for the varying turbulence parameter. Therefore, the failure probability at the landward toe is underestimated by a factor 3.6 when the additional load at the landward toe is not taken into account and the turbulence parameter is kept constant along the dike profile.

### 4.2. Additional Load at Damaged Spots

The markers in Figure 9 show the combination of the turbulence parameter and critical velocity leading to failure of the dike cover for the eight test sections (Table 3). A linear fit through these failure points results in a relation for the turbulence parameter at damaged spots

$$\omega = 0.074\, U_C + 2.1 \tag{6}$$

with a root-mean-square error of 0.14 and a coefficient of determination $R^2$ of 0.56. The constant 0.074 has the units s/m to ensure that the turbulence parameter is dimensionless.

The calibrated relationship results in a turbulence parameter of 2.56, 2.38, 2.26 and 2.15 for a good grass cover, average grass cover, poor grass cover and clay cover, respectively, using the values for the critical velocity in Table 2. The calibrated turbulence parameter for additional load at damaged spots solely depends on the critical velocity, but the total load in the model is described by the turbulence parameter and the flow velocity. This means that the total load increases when the flow velocity increases and is therefore higher for damages on the lower slope compared to the upper slope.

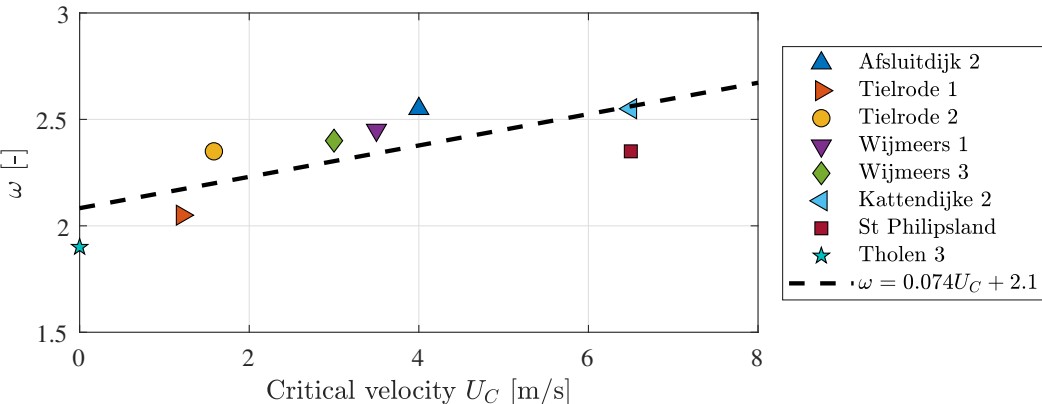

**Figure 9.** The relationship between the turbulence parameter $\omega$ and the critical velocity $U_C$ based on a linear fit through the failure points of the eight test sections.

### 4.3. Failure Probabilities of Damaged Spots

The conditional failure probabilities $P_{f|h}$ and the total failure probabilities $P_f$ are compared for failure at the landward toe and failure at damaged spots along the slope and for four cover qualities: clay, poor grass, average grass and good grass (Figure 10). The fragility curves show the conditional failure probability as function of the free crest height $R_c$ and they do not reach 1 for the cases with a grass cover (Figure 10a). This is related to the crest height of 17.93 m+NAP and the maximum water level from the HydraNL software of 17.65 m+NAP, which means that no information is available for free crest heights smaller than 0.3 m where most of the wave overtopping occurs. However, these water levels have a return frequency smaller than $10^{-6}$ yr$^{-1}$ and their contribution to the failure probability are therefore negligible. Significant wave overtopping occurs for the computed water levels with a maximum average overtopping discharge of 250 L/s/m and an average

of all storm events of 20 L/s/m for a water level of 17.65 m+NAP. Wave overtopping tests on grass-covered dikes showed that good grass-covers can easily withstand an overtopping discharge of 20 L/s/m [20] which agrees with the small conditional failure probability for good grass covers.

The higher conditional failure probabilities for the clay cover are the result of a lower critical velocity and a larger inverse strength parameter. The lower critical velocity means that small waves with low velocities are able to erode the cover. Additionally, the waves erode more material due to the large inverse strength parameter so the erosion depth of 20 cm is already reached with little overtopping. This results in a fragility curve where the conditional failure probability reaches 1 at a free crest height of 0.6 m which corresponds to an average overtopping discharge of 3 L/s/m.

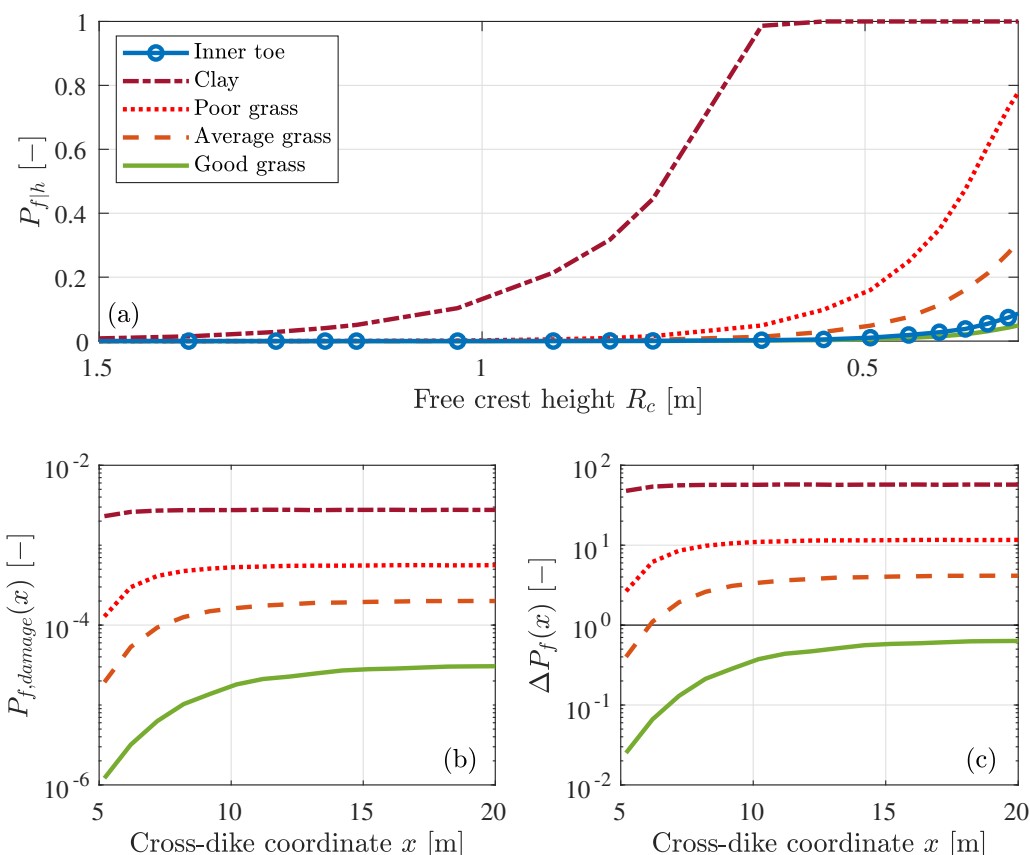

**Figure 10.** (**a**) The fragility curves showing the conditional failure probability $P_{f|h}$ as function of the free crest height $R_c$ for failure at the landward toe and failure at a damage around 10 m from the crest for four cover qualities. (**b**) The cross-dike failure probability for a damaged spot on the slope $P_{f,damage}(x)$ for the four cover qualities. (**c**) The ratio $\Delta P_f(x)$ for the four cover qualities (Equation (5)).

The failure probabilities for a dike cover with a damage are compared to the failure probability at the landward toe using their ratio $\Delta P_f(x)$ (Equation (5)). The results show that a dike profile with a damaged spot is more likely to fail compared to an undamaged dike profile at the landward toe with a good grass cover, except when the grass cover at the damaged spot is still intact and of good quality. The ratio $\Delta P_f(x)$ is always smaller than 1 for damages with good grass cover (Figure 10c), which means that the dike is more likely to fail at the landward toe than at the damaged spot. An example of such a damage is a small slope instability where the grass cover is still intact, but the small cliff leads to an additional load at this location. The calibrated turbulence parameter for a good grass cover ($\omega = 2.56$) is smaller than the turbulence parameter at the landward toe ($\omega_{toe} = 2.75$)

leading to a lower failure probability at the damaged spot. Additionally, the load on the slope is lower compared to the landward toe because of a lower flow velocity.

The lower flow velocity on the slope is also the reason why the ratio is smaller than 1 for an average grass cover at the beginning of the slope, but the ratio increases along the slope to a ratio of 4. The ratio is especially large for damaged spots with a bare clay cover. The results show that a dike profile with only clay as cover is 60 times more likely to fail than a regular dike profile.

## 5. Discussion

In this study, we developed a framework to calculate the probability of dike cover failure by overtopping waves. The variation in the load along the dike profile is included in the GEM model using a turbulence parameter and the flow velocity along the profile. This developed framework has two innovative applications: it enables us to calculate (1) the cross-dike failure probability to identify vulnerable locations for dike cover failure, and (2) the failure probability of damaged dike profiles. Both applications are discussed together with the method to derive the additional load at the damage and a comparison with other studies.

### 5.1. The Cross-Dike Failure Probability

The cross-dike failure probability indicates that the landward toe is most vulnerable location for dike cover erosion due to the high flow velocities at the end of the slope. In case the turbulence parameter for the landward toe of Frankena [34] is used, the dike profile is 150 times more likely to fail at the landward toe compared to the upper slope.

The cross-dike variation in the hydraulic load is included in the model using the maximum flow velocity along the dike profile and the turbulence parameter. The turbulence parameter can be used to account for the additional load at damages and transitions and thereby the framework can also be applied to other irregularities in a grass-covered dike profile, such as transitions and multi-functional dikes [13,15,49]. This requires information on how these elements affect both the hydraulic load and the cover strength. Theoretical load and strength factors for transitions are derived by Van Hoven et al. [35] for geometric transitions, revetment transitions and vertical objects, but they recommended to validate these factors in more detail using prototype experiments. Warmink et al. [19] developed a method to calibrate these load factors and showed that the calibrated load factors are not comparable to the theoretical factors for a slope change. Detailed numerical models [5,22] can be used to determine the additional load at transitions and develop load factors such as the turbulence parameter that can be used in erosion models. Additionally, more measurements of the turbulence intensity along the profile are necessary to determine a relation for the cross-dike variation in load along damaged dike profiles and multi-functional dikes.

Kriebel [50] performed an sensitivity analysis on the GEM model for failure during a storm event and showed that the velocity on the crest and the critical velocity have the largest effect on the model results. However, the uncertainty in the velocity on the crest is small [11,50] and the uncertainty in the critical velocity is accounted for using a distribution. Therefore, [50] identified the inverse strength parameter and the turbulence parameter as the two parameters that have the most impact on the model results since literature reports a wide range of possible values of both parameters. The inverse strength parameter differs a factor 50 between grass and clay (Table 2) which is the main reason for the spread in the literature values. The values of Verheij et al. [36] are used in this study since the values were also used to calibrate the relation between the critical velocity and threshold velocity, where Frankena [34] showed that this combination of the threshold velocity and inverse strength parameter are able to accurately simulate the measured erosion depth during several wave overtopping tests on grass-covered dikes in the Netherlands and Belgium. The reported range of the turbulence parameter is 2.00–2.75 (Table 1) and the difference between the lower limit and upper limit of the turbulence parameter on the failure probability can be seen in Figure 8. The failure probability at the landward toe

($x = 21.4$ m) increase with a factor 3.5 when the turbulence parameter is increase from 2.00 to 2.75 in case of $w = f(x)$. This effect on the failure probability is smaller compared to the effect of grass quality where the failure probability increases by a factor 1000 from poor grass to good grass. The maximum flow velocity along the dike profile was not included in the uncertainty analysis since Van Bergeijk et al. [21] showed that the flow velocity can be accurately computed for a wide range of flood defences with their analytical formulas.

This study is limited to small damages with a maximum depth of 20 cm because the GEM solely describes scour erosion. This erosion model is able to accurately predict the erosion patterns during wave overtopping experiments on grass-covered dikes [16,22]. However, in case of larger damages with a higher cliff, other erosion mechanisms become dominant such as head-cut erosion [17,39]. These follow-up mechanisms need to be considered when the framework is extended to larger damages, which was outside of the scope of this study. Moreover, the erosion depth during a storm in the GEM is computed without a feedback between the erosion of previous waves on the flow of the next overtopping wave. Erosion holes lead to an increase in the hydraulic load and affect the overtopping flow and erosion downstream. The additional load is included in the model using the turbulence parameter, but the effects on the flow and erosion downstream are neglected in this study since we assumed that the dike would fail at the damage. The effect of erosion holes on the overtopping flow and erosion downstream is not fully understood at the moment and needs to be determined before these effects can be included in this framework. A possibility is to study the effect of erosion holes in detailed overtopping tests or with numerical models [5,22].

The framework is developed for river dikes where a constant water level during a storm is a good approximation. This approximation also holds for dikes at a lake, however, higher waves are expected at lake dikes due to a longer fetch length [51]. The framework is also applicable to sea dikes, but in this case the development of the hydraulic conditions-driven by a combination of the storm and the tides–needs to be included. Kriebel [50] showed how this framework can be extended to storms on sea dikes [52].

The framework is applied to a grass-covered river dike in the Netherlands and can be applied to flood defences globally. The water level and wind speed distribution are site specific and need to be adapted for other locations. The strength of the dike cover is described by a distribution for the critical velocity that is known for different clay and grass qualities [15,36]. Additionally, wave overtopping tests have been performed in Asia to determine the critical velocity of tropical grass species such as Bermuda, Carpet and Manilla [10,53] and therefore the framework can also be applied in these regions.

*5.2. The Effect of Damages on the Failure Probability*

The effect of damages on the landward slope depends on the type of damage and the remaining cover quality. In case of a damage where the grass cover is still intact and strong, the dike is most likely to fail at the landward toe. An example of such a damage is a small slope stability as depicted in Figure 1b where the grass cover is still intact. However, most damages to the dike profile result in weakening or removal of the grass cover. In these cases, the damaged dike is more than 4 (average grass), 12 (poor grass) or 60 (clay) times likely to fail compared to a regular profile. Damages to the grass cover are often the result of animal burrows such as mice, moles and rabbits [54] that reduce the cover strength. Therefore, these animal burrows are vulnerable locations for failure by wave overtopping. Other types of damages that are representative for this model study are erosion holes formed during wave overtopping by erosion mechanisms such as gradual scour erosion, bulging or roll-up [10]. These erosion holes often result in removal of the grass cover where only a bare clay cover remains to protect the dike core for a breach. For this study, we assumed that the clay layer was of good quality. In case of a poor clay quality [36], damages with only a clay cover have a failure probability that is 125 larger compared to a regular dike profile. This means that the clay quality of the cover layer is an important variable to determine the residual dike strength. The effect of the clay quality on the erosion rate

needs to be studied in more detail before the failure definition can be extended to larger erosion depths.

The ratios between the failure probability of a damage and the landward toe can be used as a first estimates for damages on other grass-covered dikes. The dike geometry does not seem to have a large effect on the ratio, because the ratio becomes approximately constant on the lower slope. Simulations for a lake dike in the Netherlands showed similar ratios for damages with a grass cover and therefore these ratios are also applicable to other case studies with similar cover types as mentioned above. However, the simulations for damages with solely a clay cover shows that these have a higher failure probability for lake dikes due to the difference in hydraulic conditions [51]. Further research into damages with a clay cover needs to be done to determine in which cases these ratios can be applied.

The cross-dike failure probability for damaged dike profiles shows that a damage on the lower slope has a higher failure probability compared to a damage on the upper slope. In practice, a damage on the upper slope is more critical because less material needs to be eroded before the dike breaches since the upper slope is closer to the outer slope. In this study, this effect is not included because failure is defined as an erosion depth of 20 cm. However, when the failure definition moves towards a dike breach, this effect needs to be taken into account [9,17].

### 5.3. Additional Load at Damages

The additional load at damaged spots is simulated using the turbulence parameter, which was calibrated using the results of wave overtopping tests at eight grass-covered dike sections (Equation (6)). These test sections where damaged during the tests and a small cliff formed at these damaged spots resulting in the formation of a small jet that impacts landward of the damaged spot. Both sea and river dike sections at multiple locations in the Netherlands with different grass qualities (0 m/s $\leq U_C \leq$ 6.5 m/s) were used to calibrate a relation for the turbulence parameter at damaged locations. This relation is applicable to other cases with a similar cover type consisting of grass vegetation on a clay layer with a critical velocity in the range 0 m/s $-6.5$ m/s. The relation needs to be investigated further before it can be applied to other grass types or soil types, such as Bermuda grass or sand.

We increased the inverse strength parameter stepwise in the method for the calibration of the turbulence parameter, because we assumed that the inverse strength parameter is a cover characteristic and not a function of the critical velocity. This assumption does not affect the failure points, except for the failure point of Wijmeers 3 which is exactly on the boundary between poor and average grass with a critical velocity of 3 m/s. In the current method, Wijmeers 3 is classified as poor grass ($U_C \leq 3$ m/s) with $\omega_{test} = 2.4$ which would increase to $\omega_{test} = 2.7$ when classified as average grass (3 m/s $< U_C \leq 5$ m/s). Although the assumption for the inverse strength parameter leads to a significant increase for this failure point, the other failure points are not affected leading to a small change in the linear fit ($\omega = 0.071U_C + 2.1$) which only affects the second decimal of the calibrated turbulence parameter for each cover quality.

The calibrated turbulence parameter varies between 2.15 and 2.56 for the different cover types, which means that the turbulence parameter for damages is within the range 2.00–2.75 reported by Hoffmans [4] and smaller than the turbulence parameter for the transition at the landward toe (Table 1). The turbulence parameter is related to the Darcy-Weisbach friction factor $f_{WD}$ used to express the energy dissipation of the flow (Appendix B). Scheres et al. [42] derived a Darcy-Weisbach friction factor of 0.19 for high-velocity air-water flows over a grass cover corresponding to a turbulence parameter of 2.42, which is close to the calibrated turbulence parameter for average grass ($\omega = 2.38$). The cliff near the damage shows similarities with stepped spillways. Felder and Chanson [43] determined $0.1 \leq f_{DW} \leq 0.4$ for flow over stepped spillways which corresponds to $2.16 \leq \omega \leq 2.83$, which coincides with the range of the calibrated turbulence parameter.

The calibrated turbulence parameter for additional load at damaged spots depends on the critical velocity since the amount of bed turbulence depends on the cover type, which is

included in the model using the critical velocity. The turbulence parameter determined for small damages with a maximum vertical cliff height of 20 cm. For higher cliffs, the height of the cliff as well as the impinging angle will affect the load in the impact zone [39]. Analytical formulas for jet impact show that the normal stress [41] and pressure [55] of the jet in the impact zone increases with the impinging angle. Additionally, wave impact simulations on grass covers using a jet show that the impact pressure increases with height [56,57]. Therefore, the effect of the cliff height and the impinging angle on the load needs to be investigated further when the model approach is to be extended for larger damages leading to higher cliffs.

*5.4. Comparison to Other Studies*

Aguilar-López et al. [15] calculated the cross-dike failure probability for the same dike section that we analysed using a different hydrodynamic model [22] in combination with an emulator. However, instead of using the water level and wind speed as stochastic variables, the failure probability for an average overtopping discharge using the dike cover strength as only stochastic variable. The failure probabilities of Aguilar-López et al. [15] are of the same magnitude but slight smaller compared to our results (Table 4). However, only the crest and upper slope were included in the hydrodynamic model used by Aguilar-López et al. [15] with the highest failure probability at the end of the slope. Therefore, it is likely that the failure probabilities of Aguilar-López et al. [15] increase when the model is extended to the landward toe where the load is highest and are closer to the probabilities in this study.

**Table 4.** Comparison between the model results and the results of Aguilar-López et al. [15] for the maximum failure probability for the grass-covered dike profile of Millingen a/d Rijn without any damages.

|  | Poor | Average | Good |
|---|---|---|---|
| Our framework | $3.9 \times 10^{-4}$ | $9.0 \times 10^{-5}$ | $5.8 \times 10^{-6}$ |
| Aguilar-López et al. [15] | $8.2 \times 10^{-5}$ | $5 \times 10^{-5}$ | $\leq 10^{-6}$ |

Marijnissen et al. [13] computed the failure probability for multi-functional dikes using solely the average overtopping discharge as failure criterion. The failure probabilities by wave overtopping are not reported, but the total failure probability by wave overtopping, piping and macro-stability combined is in the same range as the failure probabilities computed in this study ($10^{-6}$–$10^{-2}$). The fragility curves for wave overtopping show a steep curve where the conditional failure probability is always 1 for small free-crest boards [13]. For comparison, the fragility curves for macro-stability and piping usually increase from 0 to 1 over a few meters of water depth, while for overtopping the increase from 0 to 1 is over less than one meter water depth. The fragility curves in this study show a steep curve similar to Marijnissen et al. [13] with an increase in the conditional failure probability over less than one meter water depth.

In our framework, each sample corresponds to a storm event and therefore the failure probability per storm event is computed similar to the failure probabilities of Aguilar-López et al. [15] and Marijnissen et al. [13]. However, annual failure probabilities are often required for the safety assessment of dikes. The distributions for the water level and wind speed are currently per storm event and need to be transferred to a distribution per year to change the failure probability per storm event to an annual failure probability. An explanation of this method is provided by Vuik et al. [12] where it is important to take the correlation between the water level and the wind speed into account.

The vulnerability of dike profiles with a slope instability for wave overtopping was also investigated by Van Hoven [17]. The head-cut erosion model [39] was applied to an overtopping test on bare clay at Delftzijl and an overflow test at a slope instability at Bergambacht in the Netherlands. Van Hoven [17] concluded that an additional crest

width of 1.5 m is necessary after an instability to be safe for overtopping with an average overtopping discharge of 1 L/s/m. The residual strength for higher overtopping discharges was not determined due to uncertainties in the head-cut model, although no significant erosion was observed for both Delfzijl and Bergambacht [17]. In this study, the failure probability for damages with solely a clay cover resulted in a conditional failure probability of 1 around a free crest height of 0.6 m in which case the average overtopping discharge is approximately 3 L/s/m. However, in cases where a grass cover is still present at the damage, the failure probability is much lower and the dike cover is able to withstand higher overtopping discharges. Van Hoven [17] did not consider the first phase of erosion, namely the failure of the vegetal cover protection where the topsoil eroded by the overtopping waves. Since most of the cover strength is in the topsoil, it is important to take this initiation phase into account in order to determine the maximum allowable overtopping discharges for grass-covered dikes with a slope instability.

## 6. Conclusions

We have developed a framework to calculate the probability of dike cover failure by overtopping waves along the dike profile. This framework enables to determine the weakest location along the dike profile for dike cover erosion. The effect of transitions and damages on the hydraulic load are included in this framework using the turbulence parameter. A relationship for the turbulence parameter at damages is calibrated and used to determine the vulnerability of grass damages for wave overtopping.

The landward toe is identified as the most vulnerable location for wave overtopping for a regular grass-covered dike profile without any damages. The quality of the grass cover has a large influence on the failure probability and can increase the failure probability with a factor 1000. Furthermore, a formulation for the varying turbulence parameter shows that transitions and damages can locally increase the failure probability and are therefore vulnerable locations.

The vulnerability of grass damages for wave overtopping depends on the type and quality of the dike cover at the damaged location. When the grass cover remains intact and has a good quality, the landward toe is the most likely location to fail. However, damages that lead to a reduction of the cover strength or removal of the topsoil result in higher failure probabilities compared to failure at the landward toe. These damages are more than 10 times as likely to fail for a poor grass quality and more than 100 times more vulnerable in case of a clay cover.

The large variation in failure probability between grass and clay covers shows the importance of including the initiation phase of cover erosion in overtopping calculations. More knowledge on the erosion process of different types of grass and clay is necessary to understand the erosion resistance and speed of the different types of cover layers. This knowledge is required before the strength of the entire cover layer can be included in the safety assessment of grass-covered dikes and the failure definition for wave overtopping can be extended to larger erosion depths. If the dike strength of the entire cover layer can be used properly in erosion models, the failure calculations for overtopping will become less conservative resulting in more cost-effective design solutions and more accurate assessment tools.

**Author Contributions:** Conceptualization, V.M.v.B., V.A.V., J.J.W. and S.J.M.H.H.; methodology, V.M.v.B., V.A.V.; software, V.M.v.B., V.A.V.; validation, V.M.v.B., V.A.V.; writing—original draft preparation, V.M.v.B.; writing—review and editing, V.A.V., J.J.W. and S.J.M.H.H.; visualization, V.M.v.B., V.A.V.; supervision,V.M.v.B., J.J.W. and S.J.M.H.H.; project administration, J.J.W. and S.J.M.H.H.; funding acquisition, J.J.W. and S.J.M.H.H. All authors have read and agreed to the published version of the manuscript.

**Funding:** This research was funded by the Netherlands Organisation for Scientific Research (NWO), research programme All-Risk with project number P15-21.

**Acknowledgments:** We would like to thank Matthijs Gensen, Guido Remmerswaal, Mark van der Krogt and Joost Pol for their valuable input on failure probabilities and slope-instabilities.

**Conflicts of Interest:** The authors declare no conflict of interest.

## Appendix A. Convergence of the Failure Probability

Figure A1 shows the convergence of the failure probability for a constant turbulence parameter of 2.0 and an average grass quality. For a small number of samples, the samples are not uniformly distributed in the domain and are therefore do not correctly consider the stochastic nature of the variables resulting in a large variation in the failure probability as function of the number of samples. From $10^4$ samples onward, the failure probability convergences and the failure probability shows a small variation as function of the number of samples. The failure probability for $2 \times 10^4$ samples only differs 0.3% from the failure probability for $10^6$ samples. Therefore, the $2 \times 10^4$ samples used in this study are sufficient to obtain convergence of the failure probability.

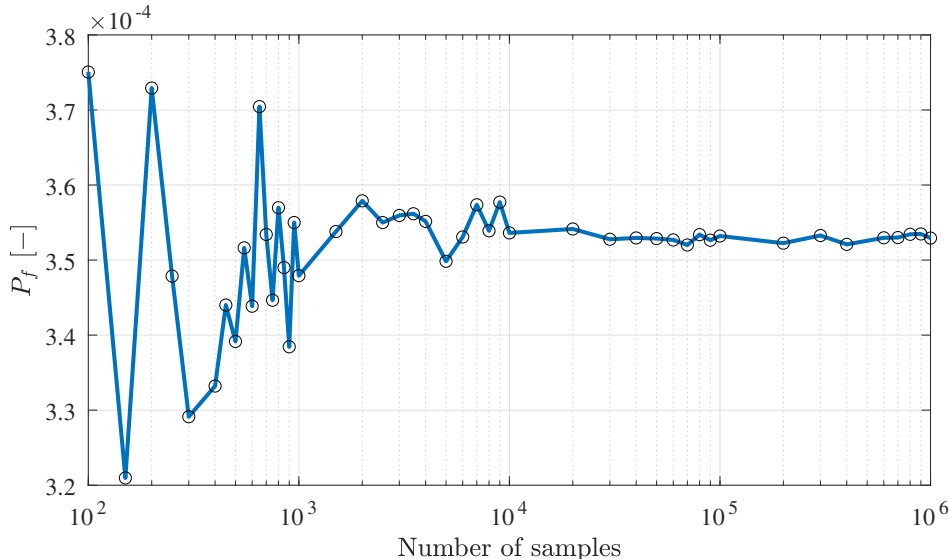

**Figure A1.** Convergence of the failure probability $P_f$ for a constant turbulence parameter of 2.0 and an average grass quality.

## Appendix B. The Depth-Averaged Relative Turbulence Intensity

Hoffmans [4] derived a formula for the depth-averaged relative turbulence intensity $r_0$ under uniform flow conditions, which was used to estimate the range of the turbulence $\omega$ for overtopping (Section 2.2).

The depth-averaged relative turbulence intensity $r_0$ is defined as

$$r_0 = \frac{\sqrt{k_0}}{U} = \alpha_0 \frac{u_*}{U_0} \tag{A1}$$

with the bed shear velocity $u_*$, the depth-averaged flow velocity $U_0$ and the constant $\alpha_0 = 1.2$. The depth averaged turbulence energy $k_0$ is defined as

$$k_0 = \frac{1}{h} \int_0^h \frac{1}{2} \left( u_{rms'}^2 + v_{rms'}^2 + w_{rms'}^2 \right) dz \tag{A2}$$

with the root mean square values of the fluctuating flow velocities in the streamwise $u_{rms'}^2$, transverse $v_{rms'}^2$ and normal $w_{rms'}^2$ directions. Under uniform flow conditions, $k_0 = (\alpha_0 u_*)^2$

resulting in Equation (A1). In uniform flow, the bed shear velocity is related to the Chezy coefficient $C$ as

$$u_* = \frac{\sqrt{gU_0}}{C} \tag{A3}$$

The Chezy formula for overtopping flow is written as

$$C = \frac{U_0}{\sqrt{h * (1 - \eta_a)S_b}} \tag{A4}$$

with the layer thickness $h*$ defined as the water depth of the wave on the crest and landward slope, the air content $\eta_a$ and the dike slope parameter $S_b = 1/\cot(\varphi)$. Combining Equations (A1), (A3) and (A4), the formula for $r_0$ becomes

$$r_0 = \frac{\alpha_0 \sqrt{g}}{C} = \frac{1.2\sqrt{gh * (1 - \eta_a)/\cot(\varphi)}}{U_0} \tag{A5}$$

For an overtopping wave, the flow velocity and layer thickness are maximum at the front of the wave [5,18] resulting in

$$r_0 = \frac{1.2\sqrt{gh_m(1 - \eta_a)/\cot(\varphi)}}{U_m} \tag{A6}$$

with the maximum layer thickness $h_m$ and the maximum flow velocity $U_m$. Hoffmans [4] used measurements of $h_m$, $U_m$ and $\eta_a$ during wave overtopping experiments on sea dikes to calculate values for $r_0$ using Equation (A6). The calculated $r_0$ varied between 0.10 and 0.25 for volumes in range of 400–5500 l/m corresponding to a turbulence parameter between 2.00 and 2.75. No clear relation between $r_0$ and the overtopping volume was found and therefore the same value of $r_0$ is used for all overtopping volumes.

The Darcy-Weisbach friction factor $f_{DW}$ is related to the Chezy coefficient as

$$f_{DW} = \frac{8g}{C^2} \tag{A7}$$

resulting in the following relation between $r_0$ and $f_{DW}$

$$r_0 = 0.42\sqrt{f_{DW}} \tag{A8}$$

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
