# Peer review of "The Cross-Dike Failure Probability by Wave Overtopping over Grass-Covered and Damaged Dikes"

_water, doi:10.3390/w13050690_

Round 1

Reviewer 1 Report

The paper deals with the application of an existing analytic model for dike slope erosion (developed by almost the same Authors) in a probabilistic fashion, i.e., by forcing the analytical model with variable input parameters so as to evaluate the failure probability along a cross-dike profile. The cases of different quality of the grass cover, and of the presence of small damages in the grass cover as well as in the dike slope, are accounted for.

Many model parameters are derived from field study at a given location and are not correctly based on the physics of the problem, so the procedure applied by the Authors cannot configure as a general framework.

Major concerns are reported below.

Major points

  1. The title of the paper should be revised. Consider the following points:
    1. failure probability (singular). The study deals with a single probability (that of failure). This probability is not “spatially distributed” in a general sense (see my last major point), but only defined along a cross-dike profile;
    2. the failure probability concerns the dikes, not the wave overtopping;
    3. the work is conducted with particular reference to NL and to a specific dike. There are serious limitations to the portability of the method to the general context (see my major point #2 below). These aspects should be reflected in the title.

I suggest: “Failure probability by wave overtopping of a grass-covered, partially damaged dike in the Netherlands”

2) Section 3.3. I have serious concerns about way of accounting for the presence of damages on the dike slope. The Authors did the job by calibrating a multiplying factor of the flow velocity, called turbulence parameter (whose physical meaning is not explicitly recalled in the text), on data collected in previous experiments. How can a multiplying factor of the flow velocity account for the effects of steps on the formation of jets and, in turn, on the erosive power of the flow? The Authors, in fact, concluded that (l. 300-301) “the calibrated turbulence parameter for additional load at damaged spots solely depends on the critical velocity”, which means that the key parameter that determine the failure is known only because, in past experiments, the flow velocity that determined the failure (i.e., the critical velocity) was measured. Such a procedure can be applied to other, possibly different cases, because there is very poor physics in this method. Calibrating a parameter out of its physical scope turns out to limit the general applicability of the procedure fundamentally.

3) Section 4.2 is not clear at all. What are the failure trends depicted in Figure 9? From which data (or which relation) have been derived the coloured curves in Figure 9? (This issue is related to the fact that Sect.3.3. is not clear as well).

To conclude, I found two main misleading issue in the presentation of the work:

  1. The work is presented as “a framework to calculate the probability of dike cover failure by overtopping waves”. The actual work is far too specific of a given location and a dike configuration to be defined as a “framework”.
  2. Speaking of a “spatially distributed” probability seems to imply a two-dimensional spatial distribution, whereas the study only deals with a cross-sectional profile. Furthermore, the failure probability profiles show nothing more than the obvious: failure occurs at the toe of the dike slope when the dike and its grass cover is undamaged, and at the partially damaged location if the initial damage is not negligibly small. Unfortunately, the assessment of the susceptibility of partially damaged locations is not treated in a sound physically-based fashion.

Minor points

-l. 21: “flow over the crest and down the…”

-l. 35: “do not include any cross-dike…”

-l. 40: “was computed…”

-l. 165: “probability”

-Table 2: missing reference in the first line of the table?

-l. 235: “velocity that lead to…”

-l. 236-ff.: Starting from the paragraph of page 8, I do not understand what is “failure trend” is actually meaning.

-l. 240: “Failure probability…”

-l. 261. “failure probability…”

-l. 272: maximum, not maximal.

-Please consider the following additional references concerning the failure of grass-covered river dikes:

Mazzoleni, M., Dottori, F., Brandimarte, L., Tekle, S., & Martina, M. L. (2017). Effects of levee cover strength on flood mapping in the case of levee breach due to overtopping. Hydrological Sciences Journal, 62(6), 892-910. https://doi.org/10.1080/02626667.2016.1246800

Viero, D. P., D’Alpaos, A., Carniello, L., & Defina, A. (2013). Mathematical modeling of flooding due to river bank failure. Advances in Water Resources, 59, 82-94. https://doi.org/10.1016/j.advwatres.2013.05.011

Reviewer 2 Report

     The work develops a probabilistic framework to assess the failure probability across grass-covered dikes in rivers due to wave overtopping, and implements it for The Netherlands usual technical context for these structures. The approach is not innovative from a conceptual point of view, but its particularization for the type of failure and dike provides further light onto the most likely location of damage in these dikes as well as the impacts of existing damage on the probability of failure.

The manuscript is well structured and its sections include the required information to understand both the methodological design and workflow, and the results and their discussion. The edition is of high quality regarding the text and graphics. Some issues, however, can be clarified in the text in order to improve its accuracy and provide better assessment on the underlying hypothesis, assumptions, and choices in the modelling of the studied failure context:

  1. In section 2, the definition of failure as an stochastic event whose probability is assessed can be more precise in terms of the time scale. Is it failure due to overtopping during a single event of overtopping, a single storm, a given period? This can be deduced by reading the work but it must be clear from the beginning.
  2. Section 2: the conditional failure probability (for a given geometry and cover quality) is defined as conditional to the water level, h, but from the scheme in Fig. 3 and the explanation it seems that the wind speed is also a conditional factor. Please, clarify this for the reader.
  3. Section 3, 3.1 Study area: please, could you include a figure with a scheme of the dike selected as study case (geometry, dimensions...)? This would really help the reader even if it can be read in the text.
  4. Section 3, 3.2 (line 196): please, comment on the potential influence of the selected value of the spatial resolution (0.5 m) on the results.
  5. Section 3, 3.2 (line 211-212): please, justify this assumption of constant value for the turbulence parameter (and the used value) and assess how well this represents the real conditions of the study case and/or under what conditions this would not hold.
  6. Section 3, 3.3 (lines 240-245): please, justify this decision.
  7. Section 3, lines 260-261: I find the use of "vulnerability" to name the indicator given by equation 5 somehow confusing, since this is usually used in risk assessment to name a conditional probability (for example, associated to failure occurrence conditioned to agent occurrence). The meaning of this ratio is clearly explained in terms of its value, my question is just on the name, is this usually used?
  8. Fig. 8: are the discontinuities in the yellow probability curve in the graph b) the result of the parameterization used for w(x)?
  9. Fig. 10: From graph c), it could be concluded that the ratio Delta Pf (x) tends to a quasi-constant value from a point on across the dike, and it shows little variation in case of no vegetation cover. Moreover, as it is stated, it is higher than 1 for most of the domains. Is this result scalable or just the result obtained for this particular dike (geometry, drivers) and erosion model? 

 Finally, the developed approach includes, as commented, several modelling decisions in the choice of equations, adoption of values, etcetera. All these have an influence on the results, and each one of them have an impact on the global uncertainty of the probability estimation. I am curious about the possibility of testing further the approach's capability by analizing some dike before and after an event from historical data, by comparing the most likely failure type and location with the observed failures. The discussion section could assess what the major sources of uncertainty in this formulation are, and how dependent the results (and the conclusions) are of the geometry and dimensions of the study dike in this work.

Reviewer 3 Report

The present paper: “Spatially distributed failure probabilities of wave overtopping over grass-covered and damaged dikes” deals with a probabilistic approach to determine the failure probability of overtopping waves on grass-covered dikes. This is an interesting topic to be considered in this Journal, but some considerations must be made in this reviewer opinion.

Abstract. Abstract is well presented, resuming the main achievements of the research. Nevertheless, in this reviewer opinion, novelty of the research must be here indicated, in order to introduce to readers on the interest in continuing reading.

  1. Introduction is clear, with references to the previous approaches of the problem and graphical representations of the real importance of this risk. Many references are described and the main goals and objectives of the work are here presented. In this reviewer opinion, it would be interesting to see which is the different between the present approach to determine the failure probabilities and those provided in the references, to remark the impact of this work. It is widely presented in discussion parts of the document, but a small remark could be done in this section.
  2. Framework for the failure probability of wave overtopping. This section presents the methodology implemented in the probabilistic determination. In this reviewer opinion, this content should be included in a Materials and Methods section, together with section 3 to be clear with the scientific method, or in the introduction if authors consider that this is a preliminary content.
  3. Methods and case study are well described. In this reviewer opinion, the justification of turbulence parameter values could be extended. Functions of probability and vulnerability are well described.
  4. Calibration process is interesting and important for giving validity to results. The proposed relation for the turbulence parameter at damaged spots is very interesting. Could be this expression extrapolated to other case studies? Does this justify the previous determination of turbulence parameter for the model?. Description of failure probabilities is also very interesting.
  5. In this reviewer opinion, discussion is well presented, determining the possibilities of extrapolation of the present methodology and possible limitations. Comparison with other previous research is also very interesting.
  6. Conclusions are well presented, considering the previous results and interesting considerations of probabilities and risks. Nevertheless, in this reviewer opinion, novelties should be remarked again in the conclusions.

Round 2

Reviewer 1 Report

See the attached pdf.

Reviewer 2 Report

I thank the Authors for their reply and revised version. Most of my comments have been clarified accordingly. However, I am not convinced in the case of two remaining issues:

(4) I am not sure what the Authors mean by "The spatial resolution has no effect on the results, since the analytical formulas are independent of the spatial resolution". The use of analytical formula in a discretized domain involves scale issues, they might be negligible but this should be assessed. Could you please clarify further? Maybe I misunderstood this sentence.

(5) In this case, taking also into account the Authors' reply to my final comment in the first report and the fact that,  as they say,  "Because the framework is based on the analytical gras-erosion model, the uncertainty in the results are mainly the effect of parameters in this model", I am not convinced about the explanation given to the cross-dike variability of the turbulence parameter being too complex to consider. This is specially relevant given the differences found when some variability is included in special transitions, as they show. Even if the computing effort was too high, which is not fully justified in the text, the order of magnitude of the uncertainty might result in this formulation not being adequate.

A minor issue is that I still wonder whether some field test on already damaged dikes could verify, at least approximately, the adequacy of the model. In any case, the title could reflect the model being specific within the national framework of technical requirements.

Round 3

Reviewer 1 Report

The paper can be accepted in present form.

Reviewer 2 Report

I thank the Authors for the clarifications and modifications done. I can accept the manuscript in this current version.

Kindest regards